# Effects of Metformin on Bone Mineral Density and Adiposity-Associated Pathways in Animal Models with Type 2 Diabetes Mellitus: A Systematic Review

**DOI:** 10.3390/jcm11144193

**Published:** 2022-07-19

**Authors:** Darren Kin Wai Loh, Amudha Kadirvelu, Narendra Pamidi

**Affiliations:** Jeffrey Cheah School of Medicine and Health Sciences, Monash University Malaysia, Jalan Lagoon Selatan, Bandar Sunway, Subang Jaya 47500, Malaysia; darrenloh.kw@gmail.com (D.K.W.L.); amudha.kadirvelu@monash.edu (A.K.)

**Keywords:** metformin, type 2 diabetes, bone density, adiposity, animal

## Abstract

Recently, there have been investigations on metformin (Met) as a potential treatment for bone diseases such as osteoporosis, as researchers have outlined that type 2 diabetes mellitus (T2DM) poses an increased risk of fractures. Hence, this systematic review was conducted according to the 2020 PRISMA guidelines to evaluate the evidence that supports the bone-protective effects of metformin on male animal models with T2DM. Five databases—Google Scholar, PubMed, Wiley Online Library, SCOPUS, and ScienceDirect—were used to search for original randomized controlled trials published in English with relevant keywords. The search identified 18 articles that matched the inclusion criteria and illustrated the effects of Met on bone. This study demonstrates that Met improved bone density and reduced the effects of T2DM on adiposity formation in the animal models. Further research is needed to pinpoint the optimal dosage of Met required to exhibit these therapeutic effects.

## 1. Introduction

According to the International Diabetes Federation (IDF) in 2019, approximately 373.9 million adults between the ages of 20 and 79 years have impaired glucose tolerance [1]. Malaysia’s National Diabetes Registry Survey (NDRS) and the National Health and Morbidity Survey 2019 also reported that 3.9 million adults aged 18 and above have type 2 diabetes, corresponding to 1 in 5 adults having type 2 diabetes [2]. Further inspection revealed that patients with T2DM experience a 50–80% increased risk of bone breakage [3]. As T2DM is considered to be a risk factor, more research is necessary to identify possible preventive measures for these patients. On the other hand, Cheung et al. [4] anticipate that the statistics of hip fractures will grow considerably in Asian countries between 2018 and 2050. The annual medical expenditure is also forecasted to reach USD 10 billion by 2050, assuming that treatment costs remain constant. These statistics are concerning when considering the prevalence of T2DM and the status of the disease as a risk factor for osteoporosis.

Bone health can be described as bones being without compromise in bone density, unlike bones during osteoporosis. Features integral to bone strength include, but are not limited, to the cortical bone’s porosity and the trabecular bone’s microarchitecture [5]. Malone and Hansen discussed how although obesity and increased risk of developing T2DM are closely linked, obesity is not the primary source of the advancement of this metabolic disease [6]. Inherited insulin resistance is also considered to be one of the many factors contributing to T2DM by altering the muscle and islet α-cells. Therefore, this brings forth an upsurge in glucose and insulin release that causes increased production of glucose by the liver and insulin by the pancreas. The quantification unit for bone health is generally bone mineral density (BMD). However, a recent epidemiological study outlined how although higher BMD has been considered an indicator of lower fracture risk, patients with T2DM have higher BMD than healthy individuals, but still have an elevated risk of fracture [7]. Unnanuntana et al. [8] pointed out that when comparing the improvement in bone density between patients receiving osteoporosis treatment and placebo, translation of the medication’s effectiveness was only 18% when considering fracture prevention. Hence, an increase in BMD does not automatically imply a reduction in fracture risk, and other possible factors can influence a patient’s risk of fracture.

Other elements that can forecast fracture risk should be noted, such as measuring serum adiponectin levels. Komorita et al. [9] disclosed a correlation between elevated serum adiponectin levels and heightened risk of fractures in patients with T2DM. Moreover, Yamamoto, Yamauchi, and Sugimoto underlined how using the trabecular bone score (TBS) proved to be a more dominant factor in verifying bone fragility in patients with T2DM [10]. On the other hand, Yu et al. described how increased marrow adipose tissue (MAT) and reduction in MAT unsaturation are connected to skeletal frailty [11]. In general, the location of MAT is in the bone marrow, and its expression of MAT originates from mesenchymal cells of the bone marrow after differentiation. The study also showed an amplified proportion of MAT in obese T2DM patients, suggesting a connection between T2DM, MAT, and skeletal health.

There are currently plenty of antidiabetic medications on the market, including but not limited to the thiazolidinedione class, sulfonylurea, and sodium–glucose cotransporter (SGLT2) inhibitors [12]. However, the American Diabetes Association (ADA) mentions that metformin remains the first line of treatment for patients diagnosed with T2DM, due to its favorable results in terms of efficacy, safety, pricing, and the upside of reducing the risk of cardiovascular incidents [13]. Exploration of metformin also showed that it could induce osteogenic effects crucial for bone formation [14]. The Diabetes Prevention Program reported that overweight participants treated with metformin displayed a substantial decline in body weight, waist circumference, and prevalence of T2DM by 31% [15]. This observed mechanism begins with the signaling of AMPK on Runt-related transcription factor 2 (Runx2), which is responsible for explicitly managing the formation of osteoblasts, which are cells crucial for bone formation. The activation of AMPK also downregulates advanced glycation end products (AGEs) and allosteric site activation of insulin receptors (IRs), reducing glucose uptake and providing a suitable environment for osteoblastogenesis which, in turn, suppresses the peroxisome proliferator-activated receptor γ (PPAR-γ) and inhibits the process of adipogenesis. This signaling pathway is crucial, as osteoblasts and adipocytes share a similar progenitor cell, known as bone marrow stromal cells (BMSCs); differentiation favoring either pathway could be a factor in contributing to bone formation or bone loss, respectively [16].

There is a paucity of literature on metformin’s effectiveness in improving bone quality in animal models. As there is increasing evidence that T2DM is a potential risk factor for osteoporosis, the interest in researching whether metformin could be used as a preventative measure or treatment is high. This systematic review aims to identify the possible beneficial effects of metformin on bone health, focusing on bone density and adiposity, as both of these factors are indicators of the bone’s wellbeing. It is within expectations that both conditions’ prevalence will increase soon. Hence, this review aims to provide an overview of metformin’s effects on bone, along with a plausible change in the management strategy for the concomitant disease that could benefit public health.

## 2. Methodology

The Preferred Reporting Items for Systematic Reviews and Meta-Analyses (PRISMA) guidelines were referred to while conducting a sizable literature search. This literature search aimed to encapsulate the effects of metformin treatment on bone marrow density and adiposity in animal models. The electronic search incorporated the most recent studies and evidence for this review while reducing the probability of the databases used: Google Scholar, Wiley Online Library, PubMed, ScienceDirect, SCOPUS, and SpringerLink. The results were filtered, and only studies published between 2000 and 2021 were picked, diminishing the prospect of involuntarily omitting older studies. Studies had to have used metformin as part of the intervention, the animal model could not have had comorbidities, and lastly, animal models with the presence of estrogen were excluded, as this influences the bone mineral density. The inclusion criteria of relevant articles revolved around the keywords and terms mentioned individually, including “(metformin AND bone AND type 2 diabetes mellitus AND animal) OR (adiposity)”, replacing bone. The full-text articles accumulated were then imported to EndNote X9.2 as a repository, where the search results obtained were exported as EndNote files. Subsequently, using the Covidence software allowed the co-authors to aid in filtering the gathered manuscripts and possible duplications, while the remaining results were screened based on their title, abstract, and relevance to this review to reduce publication bias.

The studies included were controlled animal studies using metformin as the treatment for T2DM and recording the measurements of interest, such as the animal models’ bone mineral density and adiposity at baseline and after treatment. Interventions that utilized metformin for the treatment of T2DM were included. The administration methods, e.g., orally or through injection at any dosage, were incorporated and recorded for this review. The placebo group was the only intervention used as a contrast, and the treatment goals of all the articles accumulated were conventional.

By assessing the compiled articles, the observed pathological features of T2DM in vivo were grouped in Table 1 according to how they were analyzed; they included the following.

### 2.1. Selection of Studies

The compiled titles and abstracts from the electronic searches were screened independently by two authors (D.K.W.L. and N.P.) and assessed to select suitable research papers, for which we obtained the full text. Should any difference in opinion between the two authors arise, a third author (A.K.) would step in and solve the dispute and come to an agreed conclusion. Subsequently, two authors (D.K.W.L. and N.P.) evaluated the entire document of collected articles against the inclusion criteria. When required to work out any disagreements, the third author (A.K.) acted as a mediator whenever the two review authors could not resolve their dispute through discourse. If a study did not present all essential data, we directly reached out to the authors to seek additional information and, by observation, if the same set of information was present in more than one paper, we included the paper with the most significant number of models or the most informative data. Through assessing manuscripts that met the inclusion criteria, the aforementioned features were taken into account, and are reported in the table below. The tabulated results briefly summarize the amassed documents’ similarities and differences.

Three responses are utilized as follows for each section:Yes (the phenotypic feature was present).No (the phenotypic feature was not present).Not reported (NR; the feature not reported in the model).Only significant differences from the control were recorded as demonstrating the phenotype for quantitative data, and the results were then qualitatively assessed.

### 2.2. Rodent Selection Criteria

The impact of the female hormone estrogen on bone mass, as documented by Mondockova et al. [17], reported that the insufficiency of this hormone would result in rapid bone loss. The reduction in estrogen levels also affects the regulation of bone development, such as its inability to achieve peak bone mass, maintain bone metabolism, and suppress bone loss. Finkelstein et al. [18] also demonstrated a significant decay in bone formation during the late perimenopause phase, which was shown to progress at a similar rate as during the first postmenopausal years. Zhu et al. also mentioned that leptin resistance was observed due to these hormonal changes, leading to obesity and increased risk of diabetes and bone loss [19]. Hence, articles incorporating only male animal models were included in this systematic review, so as to provide a more controlled environment.

## 3. Results

According to the keywords in the methodology section, the search yielded 3452 records. Of the 3452 records, 1019 were from Google Scholar, 427 from PubMed, 210 from ScienceDirect, 1274 from Wiley Online Library, and 522 from SCOPUS. The total number of articles removed was 3415 after filtering the records through the exclusion criteria, comprising (a) 650 articles that were not original research articles, (b) 268 articles not related to the aim, (c) 1,027 duplicates, and (d) 1,470 articles not related to the scope of the review (Figure 1). The remaining 36 articles underwent complete text analysis, and we removed 18 articles, while the remaining 18 articles were appropriate. The 18 articles selected are summarized in Table 2 and Table 3, and discussed in the present systematic review.

### 3.1. Attributes of Subjects

Through the accumulation of 18 studies, a total of 790 mice were included—not including the work of Bornstein et al. [24], where the number was not explicitly mentioned—and the age varied from 4 weeks to 18 weeks old. Furthermore, the study with the largest sample size was that of Pereira et al. [26], with 166 rats, and the study of de Oliveira Santana et al. [35] was the smallest, with 28. Regarding the strain of the mice, 5 of the 18 studies used C57BL/6 mice, another 5 used Sprague-Dawley rats, and 4 used Wistar rats.

### 3.2. Nature of the Studies

According to the tabulated data, the duration of treatment ranged from 10 days to as long as 16 weeks. Moreover, the metformin dosage also varied, from 10 mg/kg/day to as much as 900 mg/kg/day. All of the included studies were randomized controlled trials. Seven studies in this systematic review euthanatized the animals to observe both the tibia and the femur, whereas the others looked at alveolar bones, and one study investigated parietal bones. Meanwhile, for adiposity, four studies examined total serum triglyceride, three examined white adipose tissues, two examined epididymal adipose tissue, and one examined mesenteric vascular bed adiposity index.

## 4. Discussion

### 4.1. Effects of Type 2 Diabetes Mellitus on Bone Mineral Density

The relationship between T2DM and bone health has not been adequately addressed; therefore, an approach to defining bone health and the risk factors associated with T2DM patients should provide a clear overview of the ramifications. To identify treatment plans for managing these diseases, researchers must first understand the pathogenesis of the bone disease, and recognize the process of bone modeling and remodeling. Bone modeling and remodeling occur from birth to adulthood, and only remodeling occurs afterwards. During bone remodeling, osteoclasts**’** bone resorption rate is higher than their bone formation. When osteocytes discern a malformation of the bone or the emergence of microcracks, they initiate the resorption procedure, leading to the resorption of the damaged area and repairing the new bone. However, complications arise when the resorption rate increases further due to other risk factors, such as ageing T2DM, causing further reductions in bone quality and density, and disrupted microarchitecture. To date, medications prescribed for osteoporosis can reduce the risk of fractures, but with limitations, including multiple contraindications and side effects, such as hypocalcemia. Thus, healthcare professionals are faced with significant challenges when deciding suitable treatment options [38].

Multiple studies have shown that T2DM affects the process of bone turnover through the reduction of osteoid—a pre-mineralized tissue that transforms into the lamellae after calcification in the bone matrix [39]—in terms of its thickness, volume, and the osteoblast surface area when compared to individuals without T2DM of the same age [40]. Furthermore, as the accumulation of AGEs continued, diminished osteoblast formation was reported in patients with T2DM. AGEs are proteins or lipids modified through the glycosylation process in the presence of sugar; as such, observations of patients with T2DM are likely to reveal a higher aggregation of AGEs in the body [41]. The buildup of AGEs causes modifications to the features of collagen and laminin, contributing to frailty and leading to lower BMD [42]. Both of these parameters influence the generation of osteoblasts, which are derived from mesenchymal cells, and are responsible for producing type I collagen and mineralization of the bone, which then matures to form osteocytes [43].

Furthermore, as obesity increases, the buildup of visceral adipose tissue promotes the production of non-esterified fatty acids (NEFAs), consisting mainly of saturated palmitic acid that induces inflammation and insulin resistance [44]. Liu et al. [45] also highlighted how palmitic acid upregulates the transcription and translation process of PPAR-γ and spindle and kinetochore-associated protein 1 (SKA1). Their study noted that PPAR-γ was able to bolster the levels of SKA1, and found further evidence of a predicted binding site for PPAR-γ on the promoter region of SKA1 through the JASPAR database. Overexpression of SKA1 is potentially cancer-inducing, as a study conducted by Wang et al. [46] demonstrated a connection between SKA1 and the advancement of prostate cancer. Furthermore, PPAR-γ can also alter the BMSC lineage by encouraging adipogenesis through the stimulation of adipogenic pathways. This action inhibits osteoblastogenesis and the formation of osteoblasts, leading to diminished bone quality and mass. Increased bone marrow adipocytes were able to repress cells of osteogenic descent and support the production of osteoclasts from hematopoietic stem cells (HSCs) by secreting adipokines, signaling for bone resorption and, hence, losing bone mass [47].

A systematic review conducted by Adulyaritthikul et al. [21] demonstrated the effects of T2DM on the bone by utilizing male type 2 diabetic Goto-Kakizaki (GK) rats as an example of a non-obese diabetic model, and using a synchrotron computed tomography (synchrotron CT) machine to scan for changes in bone porosity and microarchitecture. The results indicated that the percentage of bone porosity of the femur cortical bone of the T2DM group was significantly higher, with a value of 26.50 % ± 2.42%, compared to the control group (5.363% ± 0.76%). On the other hand, the porosity for the femur trabecular bone of the T2DM rats was observed to be higher, with a value of 71.07% ± 2.61% compared to the control group (38.16% ± 1.84%). The tibial, cortical, and trabecular bone recordings demonstrated that the former had a higher porosity (33.37% ± 2.87%) compared to the control (6.054% ± 0.87%). The trabecular bone also exhibited higher porosity (68.88% ± 2.43%) in contrast to the control group (44.47% ± 2.35%). Lastly, the iliac bone also illustrated a higher percentage of porosity in the T2DM group (74.28% ± 2.51%) when compared with the control group (50.67% ± 3.80%). Bornstein et al. [24] recorded several isolated tibia and femur parameters after inducing obesity and T2DM by providing a high-fat diet (HFD) (60% kcal from fat) to C57BL/6J mice for 15 weeks. The parameters were measured using a microtomographic imaging system (μCT). They included trabecular bone volume fraction (BV/TV), trabecular number (Tb.N), trabecular thickness (Tb.Th), medullary area (Ma.Ar), cortical area (Ct.Ar), total cross-sectional area (Tt.Ar), and polar moment of inertia. MAT parameters through osmium tetroxide (OsO4) staining were measured, followed by μCT, comprising marrow adipose tissue volume, marrow volume, and relative MAT volume. Based on the data collected, the HFD regimen did not display a significant change in the BV/TV or Tb.N, but increased the Tb.Th and Ma.Ar which, in turn, increased the polar moment of inertia. Concerning MAT, it increased the total MAT volume. Inspections showed no substantial changes in the femur for the variables BV/TV, Tb.N, and Tb.Th, but showed a small difference in Ma.Ar, as well as in other areas, such as the cortical area, total cross-sectional area, endocortical perimeter, and periosteal perimeter. On the other hand, the H&E staining technique quantified the visceral adipose tissue (VAT) and inguinal adipose, and upon closer inspection, the HFD procedure increased the inguinal adipocyte size and the VAT adipocyte cell size.

Other articles share similar observations, such as the work of Felice et al. [23] utilizing Wistar rats fed with a fructose-rich diet (FRD) for five weeks. The bones of interest were the left femur, tibia, and second lumbar vertebra. The long bones were measured at the distal femoral or proximal tibial metaphysis section to understand the trabecular bone better. The parameters evaluated were total bone mineral content (totBMC), total bone mineral density (totBMD), trabecular bone mineral content (trBMC), and trabecular bone mineral density (trBMD), using pQCT. On the other hand, parameters such as cortical bone mineral content (cBMC), cortical bone mineral density (cBMD), cortical thickness, periosteal circumference, and endocortical circumference were also utilized to analyze the diaphysis for cortical bone using pQCT. This study also measured the parameters BV/TV, Tb.N, Tb.Th, and trabecular separation (Tb.S). However, based on the pQCT data, there were no significant differences in the distal femoral and proximal tibial metaphysis compared to the control group. The data for the diaphysis also demonstrated no substantial discrepancies between the experimental groups, which held for the other parameters, such as BV/TV, Tb.N, Tb.Th, and Tb.S. Molinuevo et al. [25] illustrated that after the introduction of T2DM, the animal models experienced a decrease in bone thickness when comparing the control group (30 ± 1) and the diabetic group (19 ± 1). Pereira et al. [26] conducted their research and discovered that the introduction of T2DM affected the alveolar bone, and was graded with a histopathological score of 3, explaining that there was complete reabsorption of the alveolar process and drastic loss of the cement. BMD decreased to a value of 0.11 g/cm^2^ in models with T2DM in the study conducted by Zheng et al. [31], compared to the control group, with a value of 0.17 g/cm^2^. Research from Zhou et al. [30] also exhibited an alveolar bone loss area of 65 × 10^−2^ mm^2^ when compared to the control group, which had a value of 57 × 10^−2^ mm^2^.

### 4.2. Effects of Type 2 Diabetes Mellitus on Marrow Adiposity

On the other hand, bone marrow adipose tissue (BMAT) is fat content commonly found within the bone marrow space, accounting for roughly 50–70% of the marrow volume during adulthood and 10% of the total adipose tissue depot in humans. These tissues were initially considered to inert and to have no functional properties, but recent studies demonstrated that BMAT could manage metabolic functions. To better understand how T2DM affects BMAT, the characteristics of BMAT should be well comprehended. The growth of BMAT commences from the terminal phalanges, followed by the appendicular skeleton, emerging in the axial skeleton, and finally expecting the bone marrow to transform into MAT during the human lifespan [48]. In cases of animals with T2DM, there have been reports of the metabolic disease having a detrimental effect on the BMAT; hence, it is paramount that the potential pathophysiological changes caused by T2DM that specifically affect BMAT are well comprehended. As observed in models with T2DM, while the levels of free fatty acids (FFAs) and triglycerides reportedly increased, the likelihood of PPAR-γ activation also increased [49]. The activation of PPAR-γ can then stimulate the adipocyte differentiation process, leading to an increase in adipocyte synthesis and, therefore, an increase in BMAT.

Further evidence suggests that circumstances that are related to bone loss—such as ageing, osteoporosis, and T2DM, along with the increased growth of BMAT in this environment—cause the BMAT to lose its brown-adipose-tissue-like attribute that is necessary to supply the microenvironment that encourages osteogenesis, thus silencing the osteogenic differentiation process [47]. During this shift in the microenvironment induced by T2DM, oxidative stress resulting from the heightened levels of pro-inflammatory cytokines and stimulation of the production of reactive oxygen species (ROS) results in a deviation of the standard cellular pathway. Muruganandan et al. [47] also explained that due to the divergence of the cellular pathway favoring the FoxO transcription factors that require ß-catenin, this action diminishes the T-cell factor/lymphoid enhancer-binding factor (TCF/LEF)-mediated osteogenic signaling in MSCs, as both of these pathways compete for the same protein substrate. As such, the bone remodeling process runs amok, whereby osteogenesis is interrupted and adipogenesis is encouraged, leading to unregulated adipose production levels, reduced bone density, and impaired skeletal health.

Alternatively, another proposed course of action as to how T2DM could affect BMAT is by inhibiting growth hormone (GH) secretion from the pituitary glands, hindering bone development and lipolysis, as T2DM and obesity are closely linked. Reducing growth hormone production can worsen obesity and create a health-threatening cycle [50]. Yang et al. [51] also concluded that patients with a high body mass index (BMI) experience negatively growth hormone secretion feedback when analyzed during the growth hormone stimulation test. In a study involving animal models, mice with genetic defects of both GH receptor and insulin-like growth factor 1 (IGF-1) had reduced bone length during observations compared to mice with only one of those insufficiencies [52]. GH is purportedly responsible for synthesizing IGF-1 in rodent chondrocytes, which signals other cells in a paracrine approach that motivates chondrocyte multiplication and endochondral ossification, which are crucial in the bone formation process.

Further investigations revealed that the paracrine IGF-1 pathway primarily impacts the trabecular bone, as was observed in experiments involving the knockout of the osteoblast-explicit IGF-1 gene, which resulted in the dwindling of trabecular bone volume, along with compromised trabecular framework and mineralization. Both GH and IGF-1 play a role in the cycle of exacerbating obesity, and exploring the 11ß-hydroxysteroid dehydrogenase type 1 (11ß-HSD-1) enzyme can provide a better understanding. The 11ß-HSD-1 enzyme is most commonly found in adipose tissues and the liver, catalyzing the interconversion of inactive cortisone to active cortisol [53]. Cortisol has the physiological response of triggering the process of lipid metabolism, which generates FFAs and glycerol, with the latter required for gluconeogenesis. However, it was assumed to adopt a lipogenic characteristic and facilitate lipogenesis when there is an overabundance of free-flowing cortisol due to the increased glucose production that stimulates insulin release. Hence, insulin was responsible for the lipogenesis observed [54].

Moreover, the GH-IGF-1–11ß-HSD-1 enzyme interaction is crucial, as both GH and IGF-1 operate as its inhibitor; therefore, any disruption in the activation of either hormone could result in increased production of cortisol, leading to a disruption in circulating glucocorticoid regulation and increasing the severity of both T2DM and obesity. Additionally, GH can promote osteoblastogenesis, and these osteoblasts then express IGF-1 through the stimulation of parathyroid hormone (PTH). In another study involving the outcomes of GH replacement therapy on PTH in patients with adult GH deficiency (AGHD), the secretion of PTH increased notably after the treatment as compared to without treatment [55], which indicates that GH could play a part in promoting PTH and further encouraging bone development. This pathway increases the yield of IGF-1, which further activates the synthesis of the receptor activator of nuclear factor kappa-B ligand (RANKL) and, in response, stimulates osteoclast production, with the purpose of breaking up and absorbing the bone for remodeling. As the quantity of RANKL starts to multiply, the ratio of osteoprotegerin (OPG) to RANKL becomes disproportionate, leading to extended bone absorption and, possibly, lower bone mass if the ratio is not maintained. Mrak et al. [56] conducted a study and found that GH plays a role in preserving the RANKL:OPG ratio by stimulating the output of OPG to revert the fluctuation. Therefore, understanding the effects of T2DM on GH and adipose tissues could be a potential area of interest for therapeutic interventions related to bone diseases such as osteoporosis and obesity.

Furthermore, one of many plausible mechanisms involved in the overproduction of marrow fat in T2DM is stipulated to be visceral fat [57]. As the glucose levels in the blood increase, this leads to an elevation in insulin production to restore the disparity. However, the continuous heightened volume of insulin has detrimental effects, such as amplifying the process of lipogenesis, which can contribute to the expansion of visceral fat [54]. In addition, due to the excessive production of visceral fat and suppression of lipolysis, a large quantity of available FFAs shift towards the liver through the visceral adipose tissue (VAT) passage channels’ portal vein. The overabundance of adipose tissues further catalyzes the release of adipokines, encouraging insulin resistance [58]. One of the many adipokines that are well studied and contribute to the condition of insulin resistance is TNF-α—a pro-inflammatory cytokine. A study conducted by Hivert et al. [59] mentioned that when there is an obstruction in the expression of the TNF-α protein, it impedes the initiation of hyperinsulinemia in murine models. However, the reverse—such as the increase in the production of FFAs and triglycerides—occurs when TNF-α is administered, which validates the role of TNF-α in the pathophysiology of insulin resistance. Esposito et al. [60] conducted a study to understand the effects of a high-fat meal on endothelial function between patients with T2DM and healthy individuals. The experiment demonstrated that individuals with T2DM have a hindered response towards L-arginine and higher TNF-α levels compared to healthy individuals after a high-fat meal, implying impaired endothelial function in the former group. As endothelial function decreased, Cersosimo and DeFronzo [61] described how the rate of transport of insulin across the capillary bed was prolonged, leading to further insulin resistance. Not only that, but TNF-α was also reported to modify the permeability of the intestinal capillaries, which could hamper the metabolic action of insulin, further aggravating insulin resistance. Further reports also indicate that elevated TNF-α can directly modify the functioning of ß-cells by provoking their apoptosis, leading to the disintegration and decline of working ß-cells and, therefore, inducing insulin resistance [62]. Prolonged insulin resistance will stress the pancreatic ß-cells as they try to secrete more insulin to combat the high blood sugar levels; however, over time, they will deteriorate and possibly lose their function, participating in the pathogenesis of T2DM and obesity, resulting in increased visceral fat and BMAT [63].

Lastly, Kim and Schafer [57] delineated how plasma leptin concentration can be considered a process that increases marrow fat in T2DM. Fat cells commonly secrete the hormone leptin, including in BMAT [64]. Leptin acts on the hypothalamus at higher concentrations and induces appetite suppression and energy output to maintain homeostasis. However, in the case of T2DM, it is suggested that an oversupply of leptin may occur, as the metabolic condition impairs the functioning of leptin receptors [65]. Over an extended period of unregulated T2DM, the body develops a lower sensitivity towards leptin, and becomes resistant to the hormone, reducing its ability to distinguish between satiety and overeating, encouraging the advancement of obesity and BMAT [66].

Additionally, in a review conducted by Hamrick and Ferrari [67], it was explained that leptin also plays a direct role in managing the process of bone formation and resorption. As the BMSCs encompass leptin receptors, their successful binding increases the production of OPG, followed by the reduction in RANKL, which results in the repression of osteoclast differentiation that is responsible for bone resorption and bone loss, thus stimulating bone formation. However, subsequently, higher amounts of leptin prompt the cell death of BMSCs, which represent bimodal-shape feedback, through which early accumulation of leptin improves bone development, but consequent elevations damage the bone formation. Few studies have looked into the effects of T2DM on BMAT, some of which had similar outcomes in their plasma profiles, such as increased cholesterol and triglyceride levels. First and foremost, Lee et al. [32] experimented on male Sprague-Dawley rats to understand the effects of T2DM on mesenteric perivascular adipose tissues and overall plasma profiles in terms of triglyceride and cholesterol. The results indicated that the instigation of T2DM with a high-fat and -fructose diet triggered an upsurge in percentages of mesenteric perivascular adiposity index for both high-fat diets for eight weeks and twelve weeks, with 1.7 ± 0.1 and 1.8 ± 0.1, respectively, as compared to the control group, with values of 0.9 ± 0.04 and 0.8 ± 0.1, respectively.

On the other hand, observations showed that plasma triglycerides increased, with values of 158 ± 16 and 166 ± 21, respectively, when compared to the control values of 66 ± 9 and 86 ± 9, respectively, further validating the effects of T2DM on adiposity. Furthermore, Molinuevo et al. [25] aimed at examining the effects of streptozotocin-induced T2DM on plasma triglycerides in rats, and cholesterol also demonstrated elevated plasma levels, with a value of 59 ± 2 for cholesterol, compared to the control group (43 ± 4). In contrast, triglycerides had a value of 62 ± 5, compared to the standard group’s value of 45 ± 4. Tolosa et al. [28] hypothesized that T2DM could influence bone metabolism by inducing hypertriglyceridemia, with a fourfold increase compared to the control group, from 62 ± 7 to 252 ± 25. The adipokine TNF-α was also a measured parameter, with values increasing from 24.7 ± 1.8 to 123 ± 9.0. The H&E stains on the BMAT also demonstrated a significant increase in the T2DM model compared to its benchmark counterpart.

Moreover, the objective of Bornstein et al. [24] was to identify the effects of T2DM on BMAT through its quantification using osmium tetroxide (OsO4), along with microcomputed tomography (µCT). The results demonstrated that the MAT volume, marrow volume, and relative MAT volume significantly increased in value due to the introduction of T2DM, compared to the control group. In addition, the researchers also explained that the conducted lipid profiling displayed a considerable difference in multiple unique lipid species in HFD murine models. Lastly, the visceral (VAT) and inguinal adipose tissue were evaluated based on their adipose size, and the HFD group illustrated an escalation compared to the control batch. Furthermore, an investigation to determine the effects of T2DM on murine white adipose tissue (WAT) by measuring the average adipocyte area utilizing ImageJ software, along with H&E staining for quantification of adipocyte size, found that T2DM significantly increases both of these parameters, validating its negative consequences on MAT [34]. The subsequent study gathered the WAT of different groups and weighed it to compute the difference as percentages in body weight. The outcome showed that the HFD group significantly rose compared to the other groups, signifying the overall increase in adipose tissues in the T2DM murine model [37]. de Oliveira Santana [35] also looked into plasma criteria such as total serum cholesterol, HDL, VLDL, LDL, and triglycerides of SWISS male mice models fed with a low-carbohydrate HFD (LCHFD) to observe the variation between the test groups, along with H&E staining of WAT to quantify the weight of WAT, mesenteric adipose tissue, and retroperitoneal adipose tissue. Based on the results obtained, the aforementioned adipose tissues displayed a significant increase in weight in comparison to the control group, while the plasma parameters—such as triglycerides, LDL, and total cholesterol—also exhibited elevation with values of 143.8 ± 16.52 from 106.5 ± 4.35, 46.52 ± 11.10 from 15.76 ± 4.69, and 174.6 ± 6.104 from 113.4 ± 7.250, respectively. H&E staining also revealed that LCHFD increased the mean diameter of adipocytes, further supporting the detrimental effects of T2DM on adiposity. Lastly, Ismail et al. [36] analyzed the serum variables of mice induced with an HFD, including cholesterol, triglycerides, LDL, VLDL, and HDL, and obtained values of 190 ± 4.4 from 118.6 ± 2.9, 195.7 ± 10.5 from 98.3 ± 6, 208.6 ± 9.6 from 57.6 ± 4.4, 38 ± 2.1 from 19.6 ± 2.1, and 23.6 ± 2.1 from 34 ± 1, respectively.

### 4.3. Effects of Metformin on Bone Mineral Density

As previously mentioned, T2DM influences glucose metabolism by instigating the production of ROS and AGE, causing oxidative pressure and provoking the redirection of MSCs towards developing HSCs and osteoclasts responsible for bone resorption [68]. While the disease advances, the PPAR-γ receptor is activated, and promotes the process of adipogenesis, increasing visceral fats in the body, leading to the accumulation of BMAT and the pathogenesis of obesity, which further exacerbates the symptoms of T2DM. Healthcare professionals commonly prescribe metformin to relieve and manage the high glucose levels found in a T2DM patients and, additionally, to reverse the increased BMD and adiposity caused by the illness. A study to identify the effects of metformin on bone mineralization in male Wistar rats showed that the administration of metformin at 200 and 180 mg/kg body weight (b.w.) per day orally, three days after the induction of T2DM for four weeks, raised the levels of bone development indicators such as osteocalcin (OCN) and total alkaline phosphatase (TALP). In contrast, the bone resorption marker tartrate-resistant acid phosphatase (TRAP) was significantly reduced compared to the untreated group. Therefore, Adeyemi et al. [20] proposed that metformin usage can maintain the wellbeing of bones in models with T2DM. Adulyaritthikul et al. [21] aimed to investigate the effects of metformin treatment on bone porosity in male T2DM Goto-Kakizaki rats. The diabetic rats were given 15 mg/kg body weight of metformin twice daily through oral gavaging for four weeks, and the microarchitectures of different sections of bone were analyzed and observed through synchrotron computed tomography and imaging. The results indicated that metformin could successfully reduce femur cortical, tibial cortical, and iliac bone porosity, where the authors concluded that the usage of metformin treatment is favorable for bone health. The study by Aljalaud [22] also found similar results—the bone thickness was significantly enlarged compared to the diabetic control group when given 150 mg/kg/day of metformin for six weeks. Felice et al. [23] hypothesized that T2DM alters the orderly conservation of bone architecture, and that metformin can inhibit that side effect. The rats were allocated 100 mg/kg/day of metformin through their drinking water for three weeks, but did not display any significant changes in BMC, BMD, or the other structural criteria previously mentioned. The authors noted that although there were no visible changes based on the analysis conducted, this does not imply that the condition of the bone is not affected by T2DM or metformin, as BMD may not be a suitable evaluation tool for assessing bone quality. Moreover, the purpose of another article chosen for this systematic review was to evaluate the effects of metformin treatment on bone tissue reconstruction in male T2DM rats fed with 100 mg/kg/day for two weeks. The results indicated that the introduction of metformin promoted an increase in bone thickness and area of reossification; the researchers concluded that metformin could promote bone recovery in T2DM rat models [25]. On the other hand, there are reports of reductions in bone loss caused by metformin in T2DM rat models with periodontal disease. The study included two groups with different dosages of metformin: one with 50 mg/kg/day and the other with 100 mg/kg/day, with the latter showing lesser degradation and destruction of the alveolar process and cement [26]. In a subsequent investigation, Sun et al. [27] aimed to assess the effects of metformin utilization on orthodontic tooth movement, defined as the process when compulsion applied instigates bone resorption on the pressure section and bone apposition on the tension section in T2DM models [69]. The route of administration was intragastric, with a dosage of 100 mg/kg/day for a month, and the results suggested that metformin can favorably repair the damage caused by T2DM by increasing the availability of ALP, reducing TRAP production, and regularizing tooth movement. Furthermore, a group of researchers mentioned that T2DM is detrimental to bone metabolism and further harms the microarchitecture of the long bone, and speculated that metformin could hinder these setbacks. The male Sprague-Dawley rats were given 100 mg/kg/day of metformin in their drinking water for two weeks, which reduced TRAP activity, thus moderately improving the bone microarchitecture [28]. Additionally, a different article intended to understand the preventive impact of metformin on T2DM models, and discovered that supplying 900 mg/kg/day of metformin also significantly increased ALP, OCN, and BMD [31]. Although Zhou et al. [30] investigated the effects of metformin on inflammatory signaling, they also observed slight alveolar bone loss and tooth movement when provided 200 mg/kg/day of metformin for 10 weeks. A study headed by Zhou et al. [29] administered 200 mg/kg/day of metformin to BKS-*Lepr*^em^2C^d479^ db/db male mice for nine weeks, demonstrating partial reversion of the alveolar and periodontal bone loss induced by T2DM. Furthermore, Bornstein et al. [24] managed to illustrate that metformin could partially rescue some areas of the bone—such as the cortical bone area fraction, which was affected by T2DM—but did not affect other areas, such as the trabecular thickness, when given metformin of 300 mg/kg/day for six weeks.

### 4.4. Effects of Metformin on Marrow Adiposity

Based on current literature, T2DM plays a part in complicating the fate of MSCs by tilting the balance between osteoblastogenesis and adipogenesis, favoring the latter. However, metformin can reduce marrow adipocytes’ numbers and size by inhibiting PPAR-γ and the formation of adipocytes caused by T2DM [70]. In this study, 500 mg/kg/day of metformin was given through drinking water for two different durations: one group for 8 weeks, and another for 12 weeks. The results demonstrated that metformin prevented the increase in the adiposity index compared to the diabetic untreated group for both durations. Therefore, the researchers concluded that metformin could prevent visceral adiposity accumulation in the T2DM rat model [32].

Furthermore, Molinuevo et al. [25] showed that metformin at 100 mg/kg/day for two weeks diminished cholesterol and triglyceride levels, which are markers of increased adipocyte formation in male Sprague-Dawley rats. A study conducted with the feeding of metformin at 100 mg/kg/day through the drinking water of the rats for two weeks was able to avert the buildup of BMAT compared to the diabetic model [28]. Moreover, Bornstein et al. [24] aspired to depict the effects of metformin on BMAT, and found that the administration of 300 mg/kg/day of metformin for six weeks reduced the accumulation of BMAT catalyzed by T2DM. Additionally, the C57BL/6 mice treated with metformin at two different dosages—10 mg/kg and 50 mg/kg daily—for 14 weeks were able to lessen the impact of T2DM on their lipid profiles by reducing the total cholesterol, LDL-cholesterol, and triglyceride levels. Moreover, the quantity and diameter of fats found in visceral fat also shrunk in the group treated with metformin [33]. The study conducted by Luo et al. [34] demonstrated that metformin delivery at 250 mg/kg/day for four weeks reduced adipocyte size and volume in ob/ob C57BL/6 mice. The study performed by Pei et al. [37] contemplated the consequences of administering 200 mg/kg/day of metformin for six weeks, and found that the weight of the WAT was notably reduced in the group treated with metformin compared to the diabetic group. Subsequently, this study used SWISS male mice models to explore the effects of oral metformin at 100 mg/kg/day on the lipid profile and adipose tissues for two months, to observe any possible development. The research demonstrated reduced adipose tissue weight and cholesterol levels, signifying the suppression of PPAR-γ and adipogenesis [35]. Lastly, Ismail, Soliman, and Ismail [36] noted that using metformin at 400 mg/kg/day for two weeks improved the lipid profile parameters, such as cholesterol, triglycerides, LDL, and VLDL, while restoring HDL levels.

## 5. Strengths and Weaknesses

As murine models are inexpensive due to their small size and lower housing maintenance compared to other animals closer to the human genome—such as chimpanzees—they are still the most preferred animal models for research [71]. Furthermore, mice have simpler physiology than other model animals, and can mimic the pathophysiology of human diseases and, in this context, T2DM [72]. Moreover, as it mimics the pathophysiology of T2DM, the effects of metformin on bone could also be observed. The selected articles included a wide range of metformin dosages, from 15 mg/kg/bw to 900 mg/kg/bw, making it easier to analyze which dosage yielded the best results.

However, a significant limitation of this systematic review is the utilization of only animal models, and the translation to human studies would mean needing to cross the species barrier, requiring much time. Another discrepancy among the articles was the route of administration of metformin. There are scarce data on the differences between administering the drug ad libitum in drinking water, by oral gavaging, or intravenously, as animal models may not drink as much of the metformin dissolved in water, affecting the actual dosage provided to the animal. Hence, this warrants further investigations of the potentially different medication routes, potency levels, and other functional characteristics between the courses of administration, as the variation in the route may lead to complications and cause other unwanted obstacles. It would also be a fair and easy comparison when the study includes force-feeding or ad libitum feeding.

## 6. Future Directions

All 18 articles discussed in this systematic review presented very encouraging results as to the beneficial effects of metformin on bone mineral density and adiposity. Through the literature search, there were some studies conducted on humans to learn more about metformin and its effects on BMD and adiposity, but there were too few studies that had done so to conduct a systematic review.

By incorporating a more detailed, time-dependent, and concentration-dependent analysis, the studies of the effects of metformin could be improved. Although higher concentrations may risk unwanted side effects such as hypoglycemia and gastrointestinal complications, they may also exhibit unknown effects that could be noteworthy to increase the library of research on metformin, as previous studies have mainly focused on T2DM, but not on BMD and adiposity. Moreover, regulatory and expression studies are vital. More researchers should explore the regulatory pathways and protein expression after metformin administration, further elucidating the mechanisms affected by metformin administration. Moreover, other therapies—such as GLP1-RA or SGLT2 inhibitors—and their potential effects on bone mineral density and adiposity could also be an interesting perspective.

A study concluded on the usage of metformin-hydrochloride-loaded poly(D, L-lactide-co-glycolide) (PLGA) demonstrated better results in managing blood glucose levels, inflammation, and bone loss found in the experimental periodontal disease model as compared to the conventional drug preparation [26]. The author also stated that using this polymeric nanoparticle—a biocompatible and biodegradable synthetic polymer—offers improved therapeutic effectiveness, consistent and drawn-out drug release, decreased toxicity, stability, and lower drug decomposition. Conducting an investigative study on the safety and toxicity profiles of the metformin variants’, along with their pharmacokinetics and pharmacodynamic profiles, could also help further enumerate the information pool about metformin.

## 7. Conclusions

In this systematic review, an attempt was made to compile and discuss the effects of metformin on bone mineral density and adiposity, and the result are simplified in Figure 2. 

The studies indicate that metformin exhibits a preventative and ameliorative effect by restoring BMD and adiposity before or after the detrimental effects of type 2 diabetes mellitus in various in vivo animal models. When administered, metformin activates the AMPK pathway, which improves insulin sensitivity, suppressing adipogenesis and creating an environment suitable for MSCs to favor osteoblastogenesis. Moreover, by restoring the homeostasis of bone metabolism, the rate of bone resorption can be reduced, as the PPAR-γ is inhibited, thus promoting bone formation and remodeling to restore the loss of bone density. The reviewed articles demonstrate the benefits of metformin on bone mineral density, such as reduced bone porosity, improved bone mineral density, increased ALP, and decreased TRAP. Meanwhile, for the benefits on adiposity, the studies exhibit decreased adipocyte size, fat mass, TG, and TC. Further studies should be conducted to ensure that metformin can act as a promising drug that relieves T2DM symptoms and osteoporosis. Through this systematic review, the effects of metformin on the associated pathways of bone mineral density and adiposity demonstrated a strong interconnection, and this prospect should be further explored.

## Figures and Tables

**Figure 1 jcm-11-04193-f001:**
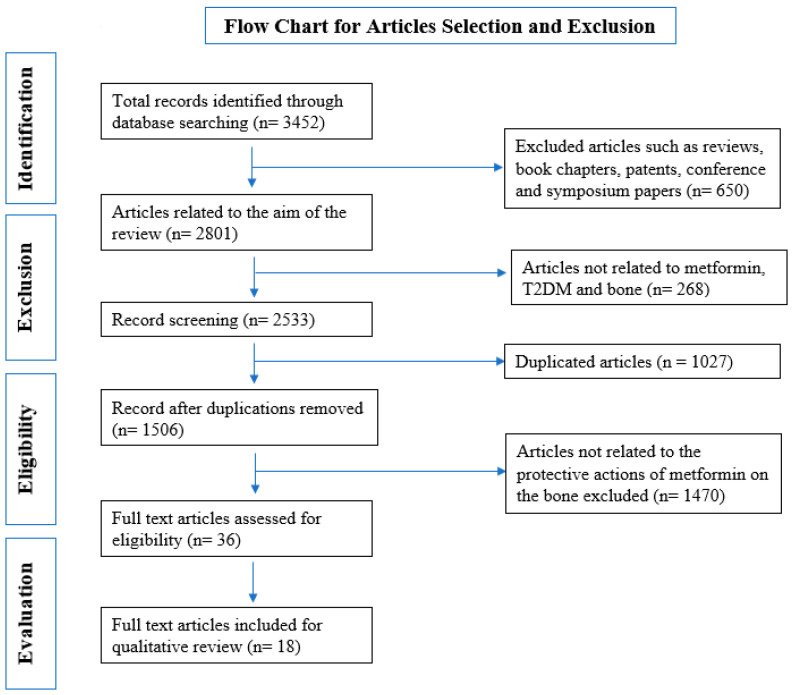
Flow chart of the study selection criteria based on the Preferred Reporting Items for Systematic Reviews and Meta-Analyses (PRISMA) guidelines.

**Figure 2 jcm-11-04193-f002:**
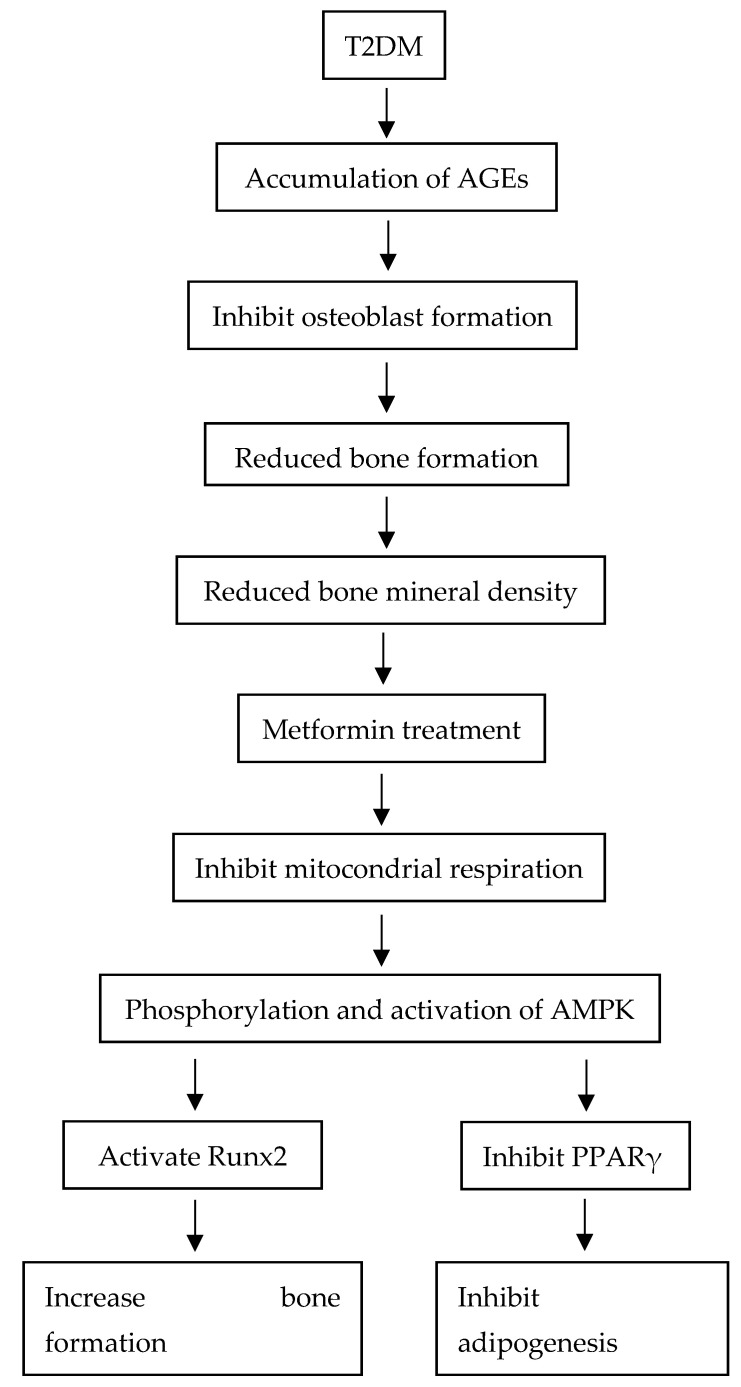
Bone pathophysiology and possible effects of metformin on BMD and adiposity [16].

**Table 1 jcm-11-04193-t001:** Features that are determinants of bone mineral density and adiposity.

Determinants of Bone Mineral Density	Determinants of Adiposity
Presence or progression of bone porosity	Decreased adipocyte size
Low bone mineral density/mineralization	Decreased fat mass
Alkaline phosphatase (ALP)	Total serum triglyceride (TG)
Tartrate-resistant acid phosphatase (TRAP)	Total cholesterol (TC)

**Table 2 jcm-11-04193-t002:** Summary of studies that reported on the effects of metformin on bone mineral density in animal models with type 2 diabetes mellitus.

First Author, Year	Mouse Strain	Age	Duration of Treatment	Dosage of Treatment	Determinants of Bone Health			
					Reduced Bone Porosity	Improved Bone Mineral Density	Increased ALP	Decreased TRAP
**Adeyemi, 2020** [20]	Wistar	10-12 weeks old	4 weeks	180 and 200 mg/kg/body weight (b.w.)/per day (p.o.)	NR	NR	Yes	Yes
**Adulyaritthikul, 2019** [21]	Goto-Kakizaki (GK) rats and Wistar (control)	NR	4 weeks	15 mg/kg b.w. twice daily (b.i.d)	Yes (cortical, trabecular and iliac bone)	NR	Yes	NR
**Aljalaud, 2019** [22]	Albino Wistar	8 weeks old	6 weeks	150 mg/kg/day	NR	Yes	NR	NR
**Felice, 2017** [23]	Wistar	8 weeks old	3 weeks	100 mg/kg/day	NR	Yes	Yes	Yes
**Bornstein, 2017** [24]	C57BL/6J	18 weeks old	6 weeks	300 mg/kg/day	Yes	Yes	NR	NR
**Molinuevo, 2010** [25]	Sprague-Dawley	NR	2 weeks	100 mg/kg/day	NR	NR	Yes	Yes
**Pereira, 2018** [26]	Wistar	NR	10 days	50 mg/kg/day and 100 mg/kg/day Poly (lac-tic-co-glycolic acid) (PLGA) + 10 mg/kg & 100 mg/kg	Yes	NR	NR	NR
**Sun, 2017** [27]	Wistar	7 weeks old	1 month	100 mg/kg/bw/day	NR	NR	Yes	Yes
**Tolosa, 2013** [28]	Sprague-Dawley	2 months old	2 weeks	100 mg/kg/day	NR	NR	Yes	No (Increase)
**Zhou, 2020** [29]	BKS-*Lepr*^em^2C^d479^ and C57BLKS (control)	6 weeks old	9 weeks	200 mg/kg/day	No (No Changes)	No (No Changes)	NR	NR
**Zhou, 2019** [30]	C57BL/6 wild-type	6 weeks old	10 weeks	200 mg/kg/day	NR	Yes	NR	NR
**Zheng, 2019** [31]	Sprague-Dawley	5 weeks old	16 weeks	900 mg/kg/day	NR	NR	Yes	Yes

**Table 3 jcm-11-04193-t003:** Summary of studies that reported on the effects of metformin on adiposity in animal models with type 2 diabetes mellitus.

First Author, Year	Mouse Strain	Age	Duration of Treatment	Dosage of Treatment	Determinants of Adiposity			
					Decreased Adipocyte Size	Decreased Fat Mass	Decreased TG	Decreased TC
**Lee, 2017** [32]	Sprague-Dawley	6 weeks old	8 weeks and 12 weeks	500 mg/kg/day	NR	Yes	Yes	NR
**Bornstein, 2017** [24]	C57BL/6J	18 weeks old	6 weeks	300 mg/kg/day	Yes	No (No Changes)	NR	NR
**Kim, 2016** [33]	C57BL/6	4 weeks old	14 weeks	10 mg/kg and 50 mg/kg	Yes	NR	Yes	Yes
**Luo, 2016** [34]	C57BL/6	8–10 weeks old	4 weeks	250 mg/kg/day	Yes	Yes	NR	NR
**de Oliveira Santana, 2016** [35]	SWISS	4 weeks old	2 months	100 mg/kg/day	Yes	NR	Yes	Yes
**Ismail, 2013** [36]	Wistar	4 weeks old	2 weeks	400 mg/kg/day	NR	NR	Yes	Yes
**Pei, 2012** [37]	Sprague-Dawley	4 weeks old	6 weeks	200 mg/kg/day	NR	Yes	NR	NR

## Data Availability

Not applicable.

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
