# Peer review of "Effects of Metformin on Bone Mineral Density and Adiposity-Associated Pathways in Animal Models with Type 2 Diabetes Mellitus: A Systematic Review"

_jcm, 2022, doi:10.3390/jcm11144193_

Round 1

Reviewer 1 Report

This interesting review focuses on the possible role of metformin on bone marrow density and adiposity. The topic is clinically relevant. The limitations of the manuscript are highlighted by the authors themselves. However, this reviewer raises some issues that need to be addressed by the authors.
1- The authors focus on the relationship between insulin resistance and TNF alpha. Notably, there is a significant relationship between the increase in TNF-alpha levels and the decrease in the endothelial function score in subjects with insulin resistance (Nutr Metab Cardiovasc Dis. 2007 May;17(4):274-9. doi: 10.1016/j.numecd.2005.11.014.). This issue with the above reference should be addressed in the discussion.
2- Metformin represents a striking example of a "historical nemesis" of a drug. About 40 years after its marketing in Europe, once demonstrated its efficacy and safety, metformin was registered also in the U.S. as anti-diabetes drug. Moreover, unpredictably up to 15-20 years ago, metformin demonstrated numerous and important extraglycemic effects. Recently several reviews have well collected the most up-to-date scientific evidence in favor of the action of metformin as an endothelial protector, as an effective drug in heart failure  as an anti-inflammatory useful in rheumatological / immunological diseases , as a beneficial drug against many aging-related morbidities (Diabetes Res Clin Pract. 2020 Feb; 160:108025. doi: 10.1016/j.diabres.2020.108025.), and finally as cancer therapy also in clinical trials. Therefore, all these issue and above references should be added in the text.
3- It would be helpful if a figure were added illustrating the beneficial effects of metformin on bone mineral density and adiposity described by the authors in the review.

Reviewer 2 Report

Loh et al. reviewed 18 articles related to effect of metformin on bone marrow density and adiposity pathway in experimental models of type 2 diabetes. Authors managed to bring novel speculations in their summary. However, I have few minor comments; 1) Although I am not a native English speaker but I can see that this paper needs extensive editing of the language. 2) Abstract needs to be improved. 3)Figure 1 is not referred in the text and Figure 2 is missing (I think authors only have one figure?), same goes for the table 1. 4)Authors are frequenlty using few words such as "following article", "another, "finding latter" article" and "next article" are bit misleading and confusing.

Reviewer 3 Report

English can be improved, and if you can add a comment regarding the therapies available for the type 2 diabetes, like GLP1-RA or SGLT2i and if there is an effect on bone marrow density and adiposity

Reviewer 4 Report

In this review, the authors investigated the effects of metformin on bone marrow density and adiposity. It is  an interesting summary, however some questions are still needed to be answered. 

1,In different model, the effects of metformin on bone marrow density and adiposity are different. The authors should discuss why the effects vary, the dose, time, or age? 

2,The author have proposed that metformin exhibits a preventative and ameliorative measure by restoring BMD and adiposity through AMPK and PPARγ signaling pathways. One graphic is needed to readers for better understanding about this review. 

3, There is short of result description in the RESULT SECTION. 

Reviewer 5 Report

Dear authors,

I have review the manuscript "Effects of Metformin on Bone Marrow Density and Adiposity 2 Associated Pathways in Animal Models with Type 2 Diabetes Mellitus: A Systematic Review”.

The aim of the present study was to evaluate the evidence  that supports the bone-protective effects of metformin on animal models with T2DM.

The results are well presented, discussed in detail.
Methods are appropriated.
The discussion section of the manuscript is well structured and comprehensive.

General comments.

A graph or figure illustrating the possible mechanisms of metformin on bone pathophysiology would be desirable.

Round 2

Reviewer 1 Report

No further comments. 

Reviewer 4 Report

The paper can be published at this form. 

This manuscript is a resubmission of an earlier submission. The following is a list of the peer review reports and author responses from that submission.